# On the gene expression landscape of cancer

**Augusto Gonzalez**[1,2], **Dario A. Leon**[2,3]*, **Yasser Perera**[4,5], **Rolando Perez**[1,6]

**1** University of Electronic Sciences and Technology of China, Chengdu, People Republic of China, **2** Institute of Cybernetics, Mathematics and Physics, Havana, Cuba, **3** Department of Mechanical Engineering and Technology Management, Norwegian University of Life Sciences, Ås, Norway, **4** China-Cuba Biotechnology Joint Innovation Center, Yongzhou, People Republic of China, **5** Center of Genetic Engineering and Biotechnology, Havana, Cuba, **6** Center of Molecular Immunology, Havana, Cuba

* dario.alejandro.leon.valido@nmbu.no

## Abstract

Kauffman picture of normal and tumor states as attractors in an abstract state space is used in order to interpret gene expression data for 15 cancer localizations obtained from The Cancer Genome Atlas. A principal component analysis of this data unveils the following qualitative aspects about tumors: 1) The state of a tissue in gene expression space can be described by a few variables. In particular, there is a single variable describing the progression from a normal tissue to a tumor. 2) Each cancer localization is characterized by a gene expression profile, in which genes have specific weights in the definition of the cancer state. There are no less than 2500 differentially-expressed genes, which lead to power-like tails in the expression distribution functions. 3) Tumors in different localizations share hundreds or even thousands of differentially expressed genes. There are 6 genes common to the 15 studied tumor localizations. 4) The tumor region is a kind of attractor. Tumors in advanced stages converge to this region independently of patient age or genetic characteristics. 5) There is a landscape of cancer in gene expression space with an approximate border separating normal tissues from tumors.

## Introduction

Nowadays, cancer is one of the biggest health problems [1]. Common knowledge relates it to aging [2], although the exact causes triggering it are unknown. There is also evidence that genetic factors play a very important role [3], specially in tumors developing in early childhood or in familial tumors. The role of external factors with carcinogenic potential is also recognized [4, 5], as it is the role of lifestyle [6]. In spite of their diversity, cancers are known to follow certain general hallmarks [7, 8].

Enormous coordinated efforts aimed at understanding basic aspects of cancer have led to many important results. A lot of information about genes, cells and tissues have been compiled and shared into public databases (see, for example, Refs. [9–11]). The analysis of such information allowed the identification of mutation signatures, immune characteristics, etc. [12–19].

**Data Availability Statement:** All the relevant data are in the paper and its Supporting information files. Moreover, the information about the data we used, the procedures and results are integrated in a public repository that is part of the project

"Processing and Analyzing Mutations and Gene Expression Data in Different Systems": \url{https://github.com/DarioALeonValido/evolp}. In particular, the PCA processing of the TCGA data is performed with the specific python scripts that are found in the folder \path{../evolp/Landscape_cancer/}.

**Funding:** The authors received no specific funding for this work.

**Competing interests:** The authors have declared that no competing interests exist.

From the theoretical point of view, there is also a great progress. We focus mainly on two particularly attractive developments. The first is the idea that cancer is an atavism, that is a cooperative state of multi-cellular organisms, prior to modern metazoans [20–24]. It means that the genetic code of cancer is implicit in our DNA and under certain circumstances it may be reestablished. The second very stimulating and general idea is Kauffman's picture of biological states as attractors in an abstract space, related to the solutions of gene regulatory networks [25, 26].

In the present paper, we use the Kauffman picture in order to interpret gene expression (GE) data for tumors. The data, provided by The Cancer Genome Atlas (TCGA) project [11], is processed by standard Principal Component Analysis (PCA) [27–29]. For a given tissue, normal and tumor samples are grouped [30]. The centers of the normal and tumor clouds of samples are identified as the corresponding Kauffman attractors. We show that the Kauffman picture may be further elaborated by stressing qualitative and semi-quantitative aspects on cancer which come straightforwardly from the analysis of GE data. Five major conclusions of this study are presented in the body of the paper. In quality of examples, let us briefly mention two of them.

First, let us stress that the coordinate along the first PC axis, PC1, which is the expression of a metagene [31], may be taken as an indicator of progression to cancer. Indeed, below we show data for kidney clear cell carcinoma (KIRC) in which it is apparent that only tumors in initial stages populate the intermediate region between normal and cancer attractors. That is, tumors in initial states show coordinates in the intermediate region whereas advanced stages of tumors correspond to coordinate values close to the center of the cancer attractor. Additionally, in a separate analysis of prostate adenocarcinoma (PRAD) [32], we show that the coordinate along PC1 correlates with clinical data on tumor cellularity of samples. In other words, the fraction of tumor cells in samples increases as we move along PC1 towards the tumor attractor.

As a second example, let us comment that for any pair of tumors we are able to compute their distance in GE space and the number of common differentially expressed genes. We show that the distance exhibits an inverse correlation with the number of common genes, i.e. the shorter the distance the larger the number of common genes. Thus, the physical distance in GE space has an actual meaning of closeness between tumors.

Besides the aforementioned 5 major conclusions, additional results are presented as Supporting Information. Other specific lines of work have been developed in separated publications, such as the analysis of PRAD mentioned above [32], the concept of smooth and abrupt transitions and their relation to GE rearrangements [33], the computation of volumes of the atractors basins and their relation to the transition rates [34], the elaboration of a 1D model for carcinogenesis based on the coordinate along PC1 [35], etc.

## Results

In this section we describe our procedures and present five main qualitative and semi-quantitative results in the form of statements. We take tissue expression data for 15 cancer localizations from the TCGA project [11], which correspond to localizations with sufficient normal and tumor samples for the PCA processing. The studied cases are shown in Table 1.

In the Methods section we explain the PCA methodology. However, let us stress two definitions which will be used in the figures below. For each gene, the differential expression in a given sample is defined as $e_{diff} = e/e_{ref}$, where $e_{ref}$ is the mean geometric average of this gene over normal samples. On the other hand, the logarithmic fold variation is defined as: $e_{fold} = log_2(e_{diff})$. This latter variable is used in the PCA results.

**Table 1. The set of normal, $N_n$, and tumor, $N_t$, samples in the 15 cancer types under study.** The TCGA notation is included.

| TCGA notation/ Cancer type | | $N_n$ | $N_t$ |
|---|---|---|---|
| BLCA | Bladder urothelial carcinoma | 19 | 414 |
| BRCA | Breast invasive carcinoma | 112 | 1096 |
| COAD | Colon adenocarcinoma | 41 | 473 |
| ESCA | Esophageal carcinoma | 11 | 160 |
| HNSC | Head and neck squamous cell carcinoma | 44 | 502 |
| KIRC | Kidney clear cell carcinoma | 72 | 539 |
| KIRP | Kidney papillary cell carcinoma | 32 | 289 |
| LIHC | Liver hepatocellular carcinoma | 50 | 374 |
| LUAD | Lung adenocarcinoma | 59 | 535 |
| LUSC | Lung squamous cell carcinoma | 49 | 502 |
| PRAD | Prostate adenocarcinoma | 52 | 499 |
| READ | Rectum adenocarcinoma | 10 | 167 |
| STAD | Stomach adenocarcinoma | 32 | 375 |
| THCA | Thyroid carcinoma | 58 | 510 |
| UCEC | Uterine corpus endometrial carcinoma | 23 | 552 |

## Statement 1

*The state of a tissue in gene expression space can be described by a few variables. In particular, there is a single variable describing the progression from a normal tissue to a tumor.*

Fig 1 shows the PCA results for three of the studied tissues: kidney renal papillary cell carcinoma (KIRP), lung squamous cell carcinoma (LUSC) and liver hepatocellular carcinoma

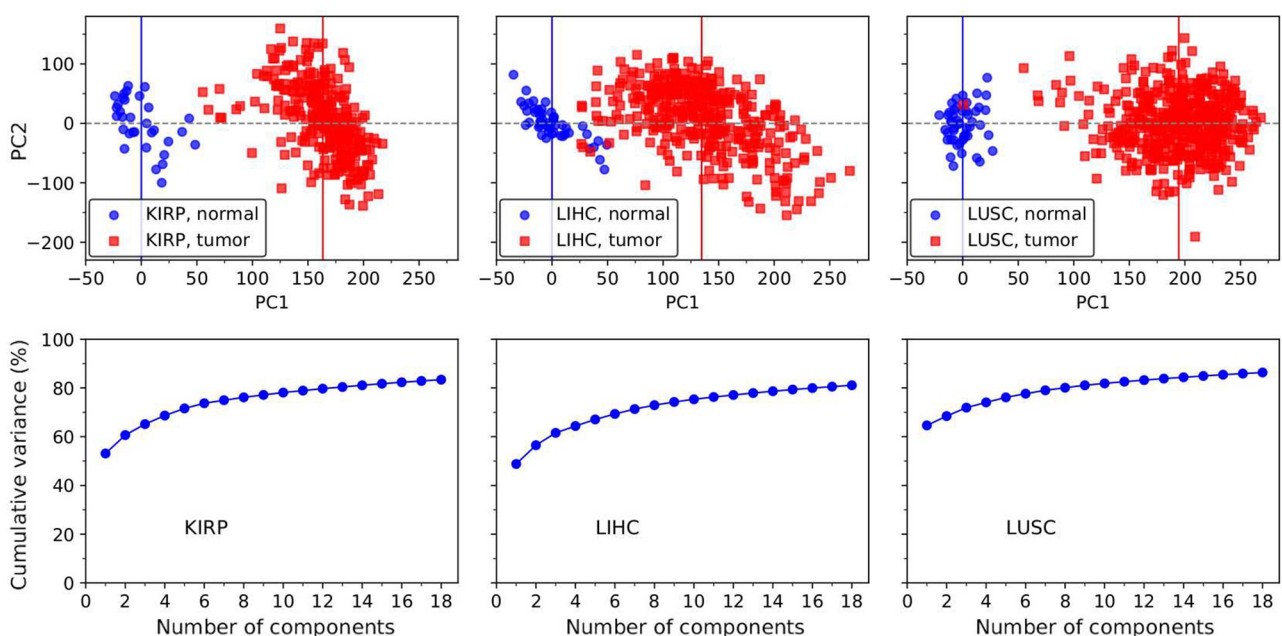

**Fig 1. Principal component analysis of the TCGA gene expression data for three of the studied tumors.** The upper panel contains the (PC1, PC2) plane, whereas the bottom panel shows the variance captured by the first 18 PCs.

(LIHC). In the upper panel, the distribution of normal and tumor samples in the (PC1, PC2) plane shows well defined normal and tumor regions, which are identified as the corresponding Kauffman attractors. Note that the first PC1 variable discriminates between a normal sample and a tumor. PC1 is thus labeled as the cancer axis and the projection along it is used to quantify the progression from a normal tissue to a tumor. Comparison with clinical variables like tumor stage or tumor cellularity is very interesting and will be discussed below.

The lower panel, on the other hand, evaluates the fraction of total variance captured by the first $n$ PC variables. In LUSC, for example, the first 3 components account for 74% of the variance. This reduced number of variables, well below the number of constituent genes, may be interpreted as the effective number of degrees of freedom of the complex system represented by a tissue.

The fraction of variance depends on the sample set. Thus, we should take the results as approximate, semi quantitative ones, that shall improve as the sample size is enlarged. Nevertheless, in spite of the statistical origin of the new variables, we can use them to describe the actual state of a given sample. Notice that each variable in fact defines an expression profile, a concerted variation of a group of genes, a metagene [31].

## Statement 2

*Each cancer localization is characterized by a gene expression profile, in which genes have specific weights in the definition of the cancer state.*

Let us call $\mathbf{v_1}$ the eigenvector along PC1 (boldface denotes vectors). We showed above that the PC1 axis accounts for a very large fraction of variance and that the projection along it can be taken as an indicator of progression towards the malignant state. For a given sample with fold expression vector $\mathbf{e_{fold}}$, the projection $x_1$ over the PC1 axis is precisely defined as

$$x_1 = \mathbf{e_{fold}} \cdot \mathbf{v_1} = \sum_i e_{fold}^i \, v_1^i. \tag{1}$$

Where the sum runs over genes, and $\mathbf{v_1}$ is the unitary vector along PC1. The $\mathbf{v_1}$ vector may be thought to provide a metagene [31] or gene expression profile of cancer in the tissue, i.e. the set of over- or under-expressed genes (and their relative importance) that define the cancer state. Each component of the normalized vector $\mathbf{v_1}$ defines the amplitude $v_1^i$ weighting the gene $i$ in the cancer state. A positive or negative sign indicates that the gene is over- or under-expressed respectively in the tumor.

The left panels of Fig 2 show the 30 genes with the largest contributions to $\mathbf{v_1}$ in the same tumors represented in Fig 1. In principle, because of their large weights, these 30 genes could be used as cancer biomarkers, however their specific roles deserve a careful study in each tissue. In LUSC, for example, the gene with the largest weight is Surfactant Protein C (SFTPC), a silenced gene with an important role in lung homeostasis [36, 37]. The analogous genes in KIRP and LIHC are Uromodulin (UMOD) and Cytochrome P450 family 1 subfamily A member 2 (CYP1A2), respectively. The ranking of genes offered by PCA for each tumor is promising and, to the best of our knowledge, have not been sufficiently exploited so far. They are hub genes [38] showing high absolute values of the differential expression, high frequencies of dysregulation in the tumor set of samples, and interactions with a large number of relevant genes. In Ref. [32] we performed an analysis of the top 33 genes in the ranking for PRAD. Some of

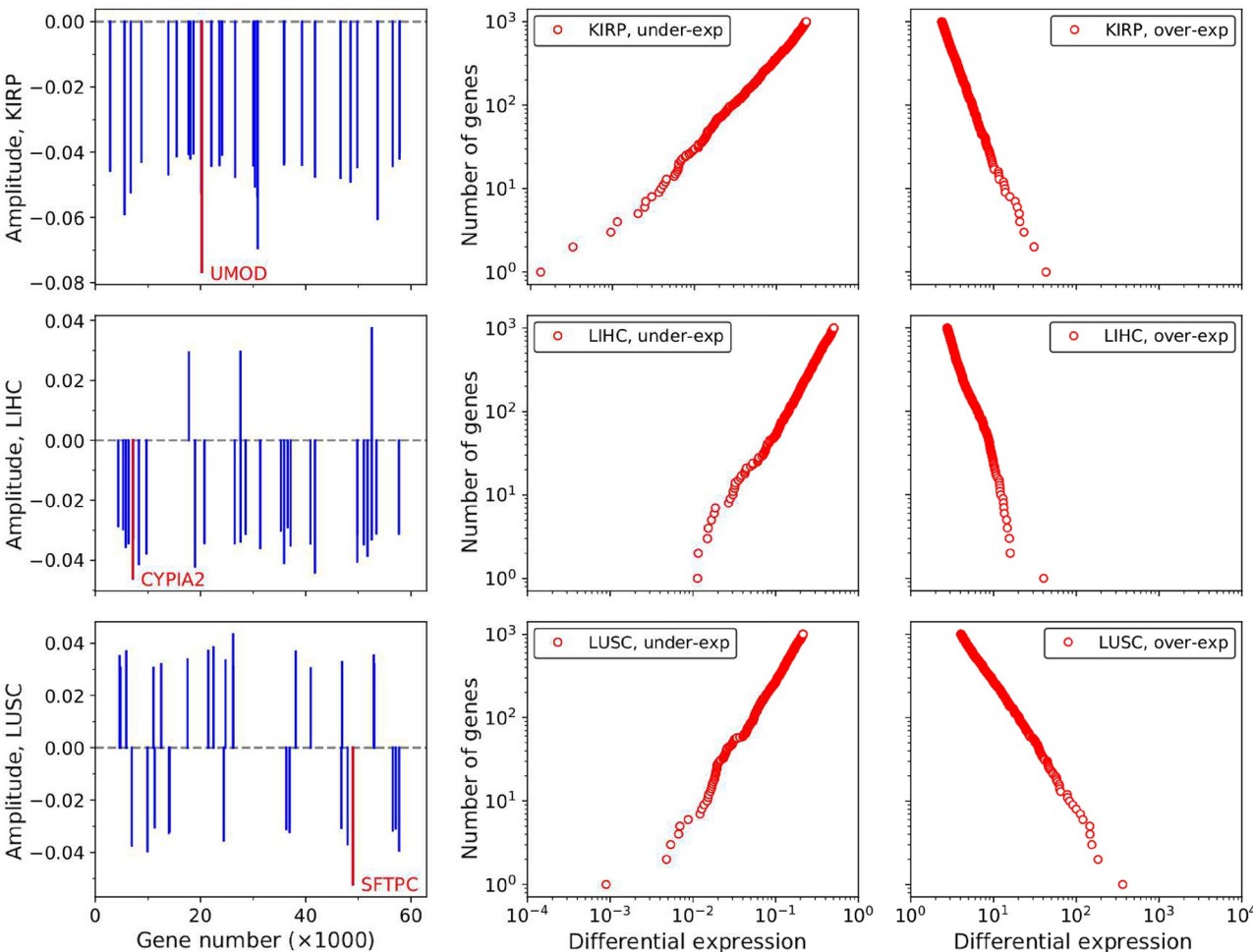

**Fig 2.** Left panel. The 30 genes with highest weights in the $v_1$ vector. The same tumors as in Fig 1 are used as examples. The numbering of genes is the one used in the TCGA data. Positive signs correspond to over-expressed, and negative to under-expressed genes. Central and right panels. Over- and under-expression tails in the integrated distribution function of genes at the center of the tumor cloud. For each gene we compute the mean geometric average over tumor samples. The over-expression tail, for example, is then obtained as the number of genes for which the average differential expression is greater than a given value.

these genes have been already validated in the literature, but there is also a number of promising, yet not validated, indications for biomarkers or target genes.

The central and right panels of Fig 2 show the gene distribution functions for the centers of the tumor clouds (geometric averages over tumor samples). In these figures, the genes are ordered according to their differential expression values. Only the over- or under-expressions tails are shown. Notice that the tails involve a few thousands of genes, the rest of the nearly 60000 RNA and protein-coding measured genes are not differentially expressed. The distributions are not symmetrical (see S1 Fig in S1 File). Whereas KIRP is dominated by silenced genes, LUSC has nearly equal proportions of under- and over-expressed genes, and in LIHC the over-expressed genes are more numerous. The log-log plots in the center and right panels of Fig 2 stress that the tails exhibit a power-like (Pareto) dependence [33, 39, 40], i.e. the number of genes with differential expression greater than a given value is proportional to an inverse power of the expression.

### Statement 3

*Tumors in different localizations share hundreds or even thousands of differentially expressed genes. There are 6 genes common to the 15 tumor localizations.*

For each localization, we select the most significant 2500 genes with the largest contributions to the vector $\mathbf{v_1}$ along the PC1 direction defining the cancer axis. This number, 2500, is roughly the number of genes with significant differential expression values and great importance in the definition of the cancer state, as it is apparent from Fig 2 central and right panels (see also S2 Fig in S1 File).

Fig 3 shows the number of shared genes for pairs of tumors. Notice that these numbers vary in the interval between 314 and 1889. Large numbers of shared genes are characteristic of tumors in the same organ but originating in different cells (lung, kidney). However, there are also tumors sharing unexpectedly large numbers of genes. For example, tumors in the uterine corpus (UCEC) and bladder (BLCA) share more than 1300 differentially expressed genes. It is worth noticing that the number of shared genes seems to be related to the proximity of tumors in the expression space (see S3 Fig in S1 File).

Let us stress that there are 49 genes common to a group of 11 tumors, BLCA-BRCA-COAD-ESCA-HNSC-LUAD-LUSC-PRAD-READ-STAD-UCEC, and six genes common to all of the studied localizations (see S1 Table). The common (pan-cancer) genes are MMP11 (+), C7 (-), ANGPTL1 (-), UBE2C (+), IQGAP3 (+) and ADH1B (-). The signs added in parenthesis indicate that the gene is over- or under-expressed in tumors. Their absolute differential expression values are very similar in all of the studied tumors.

The six identified pan-cancer genes have been recently pointed out as playing a significant role in many cancers [41–46]. It is noteworthy that these genes are straightforwardly related to cancer hallmarks [7, 8]: i.e. invasion, suppression of the immune response, angiogenesis, proliferation and changes in metabolism. Shared genes among groups of tumors suggest the possibility of global therapies in these groups. Below, we notice that pan-cancer genes play a role in both, tissue differentiation and in the definition of the border between normal tissues and tumors.

### Statement 4

*The tumor region is a kind of attractor. Tumors in advanced stages converge to this region independently of patient age or genetic characteristics.*

As shown in Fig 1, regions corresponding to normal and tumor samples are well defined and partially disjointed in the expression space. The sample dispersion comes from genetic differences, the age of patients and the evolution history of each individual. Well defined regions in expression space support the attractor paradigm of cancer [25, 26, 47], in which the cancer region should be the region of confluence of all somatic evolution trajectories which leave the normal area. In a very reductionist view one may think, for example, about the normal homeostatic state and cancer as two stable solutions of a global gene regulatory network [48, 49].

In order to test more precisely the attractor hypothesis, we study the dependence of the gene expression distribution functions on patient age. In particular, we want to check whether the distribution functions of tumors in the cancer region is age independent, i.e. whether the advanced tumor reaches in average a unique distribution function regardless of the somatic evolution history.

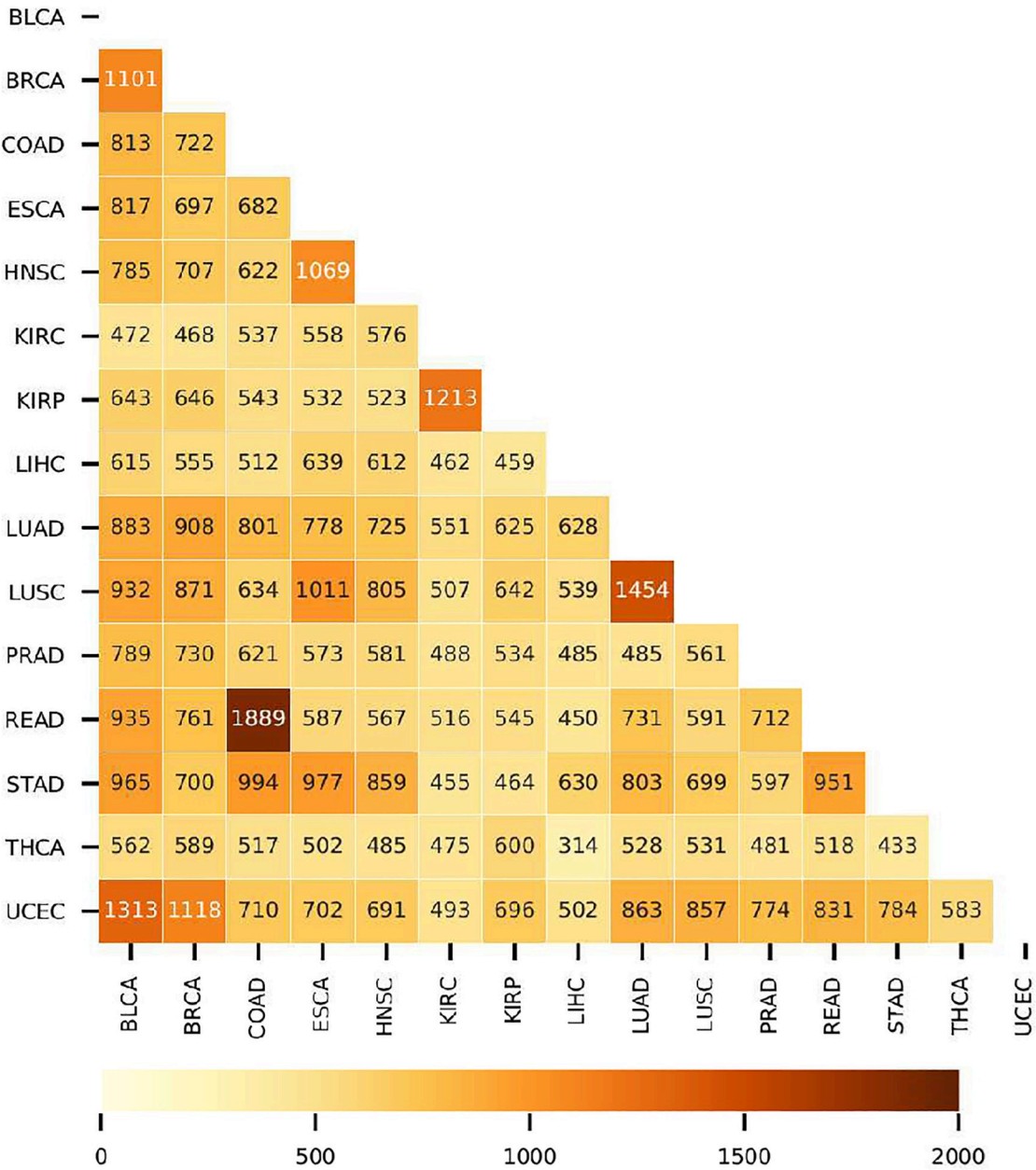

**Fig 3. Heatmap representation of the number of common differentially-expressed genes in pairs of cancer localizations.** The intersection of the vertical and horizontal coordinates shows the number of shared genes. The acronyms for the cancer names are the same as in Table 1.

Let us consider, again, LUSC as an example. According to age, we may classify the samples as young or old, defining 4 subgroups of samples in normal or tumor states: Normal Young (NY), Normal Old (NO), Tumor Young (TY) and Tumor Old (TO). These subgroups are in some sense arbitrarily defined with a threshold age of 62 years, the median of the sample set.

The results are very interesting. The (over-) expression distribution functions are visualized in Fig 4. We compute (mean geometric) averages over the NY, NO, TY and TO subgroups, and use the NY values as references in order to define normalized (differential) expression

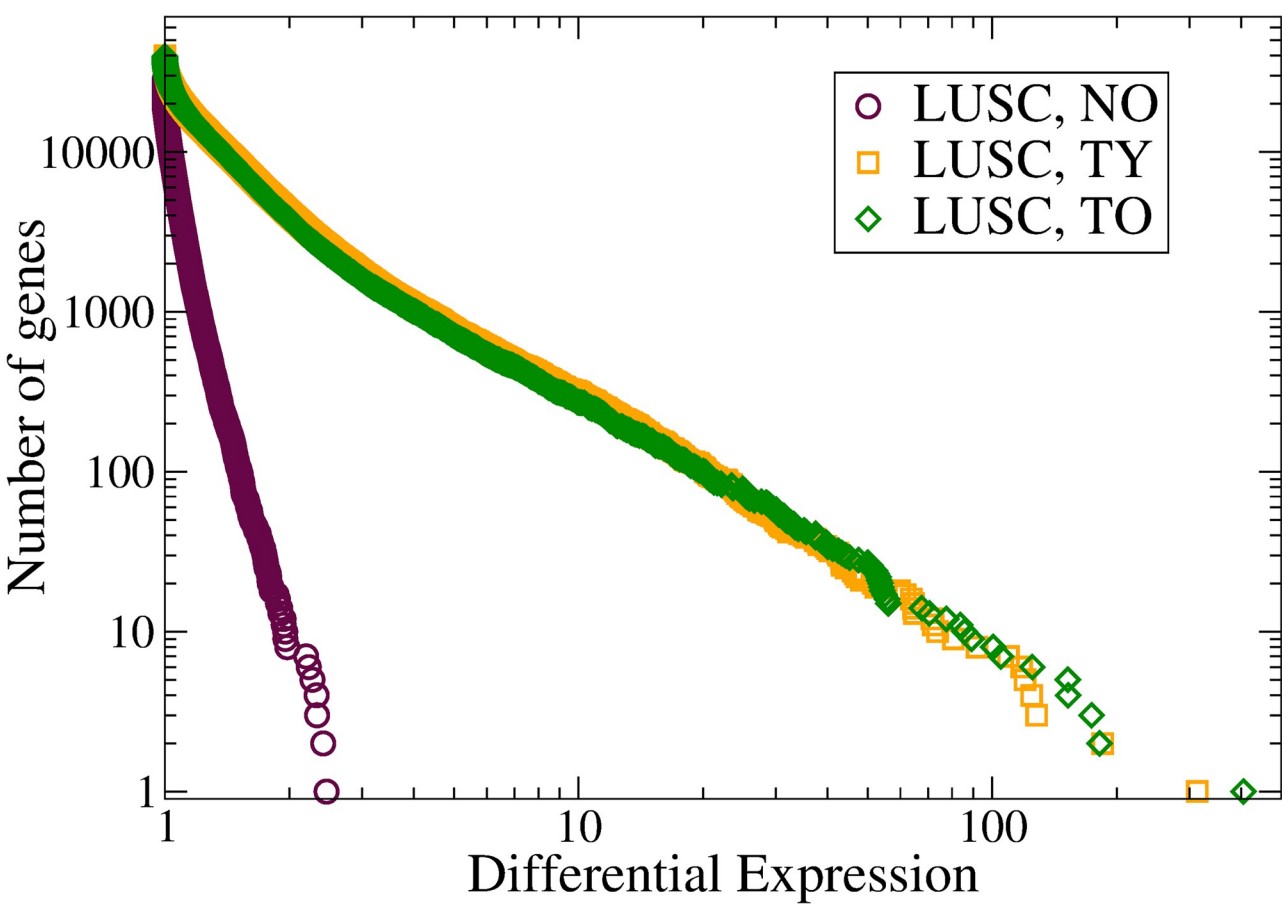

**Fig 4. Integrated gene (over-) expression distribution functions in LUSC.** According to age, samples are grouped into four sets: Normal Young (NY), Normal Old (NO), Tumor Young (TY) and Tumor Old (TO). The average over the NY set is used to define reference values in order to normalize the expressions. Each set of points represents the average over the respective group.

values in the remaining subgroups: $\mathbf{e_{diff}}$(NO), $\mathbf{e_{diff}}$(TY), and $\mathbf{e_{diff}}$(TO). These vectors characterize the centers of their respective clouds of samples. Genes are sorted with regard to their normalized expression values.

In the NO group, only a reduced number of genes, around 10, exhibit differential expression values above 2 as a consequence of aging. In tumors, however, deviations are much larger, for instance there are around 1000 over-expressed genes with $|e_{diff}| > 5$.

More striking is the similarity between the distribution functions in the TY and TO subgroups. That is, for tumors the distribution function in the final state is nearly independent of the age at which tumor initiates. This is an argument in favor of the attractor hypothesis. Similar results (not shown) are obtained for the under-expression tail.

A slightly different test comes from considering a second "time" or progression variable: the clinically determined tumor stage [50]. It is a qualification given to the tumor at the moment of diagnosis, but in some sense it quantifies also the somatic evolution once the portion of the tissue acquires the tumor condition. Fig 5 shows the distribution of tumors by stages in KIRC. Normal tissues are represented by blue points, whereas tumors are drawn in red. The four panels refer to the four stages: I, II, III and IV. All the blue points are present in the four panels, but only red points with the corresponding stage are included in each panel. We use a contour plot with a blue-to-red gradient scale of colors in order to visualize the

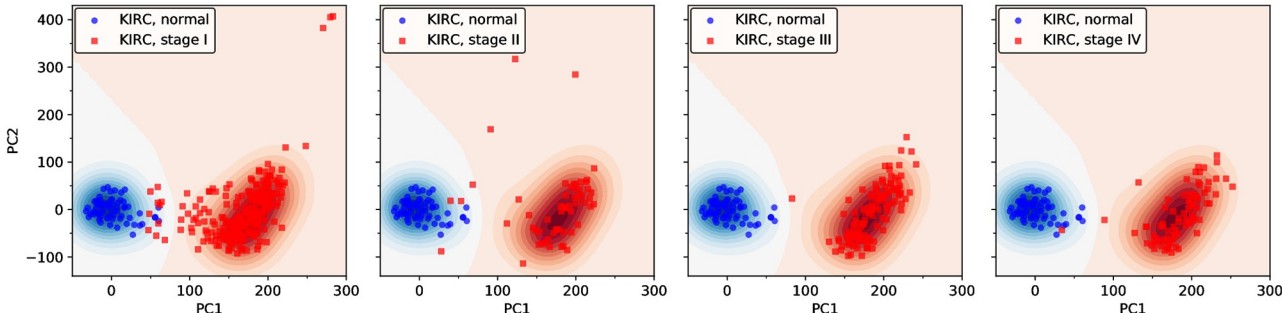

**Fig 5. Stages in the evolution of tumors in clear cell kidney cancer (KIRC).** Blue points are normal samples (included in the four panels), whereas red points are samples from tumors in a given stage of evolution. Contours represent the difference between normal and tumor density of sample points. Stages I and II seem to be "transitional", there are many points traveling along the intermediate region. On the other hand, stages III and IV are "final", in the sense that most of the tumors are concentrated in the high intensity red region.

density of points difference, $\rho_n - \rho_t$, between normal and tumor samples in the state space. Two regions of high intensity are apparent in blue and red, corresponding to normal and tumor clouds respectively, while a light zone in between indicates a small modulus density difference.

Naively, one expects that tumors move along the transition region from the normal to the tumor region as the stage evolves from I to IV. In the actual measurements, we don't track individual tumors as function of stages, but get pictures of different tumors at different stages. Thus, in the initial stages we should observe a fraction of red points captured in the transition region, whereas in the final stages most tumors should be concentrated in the optimal region. This is what actually follows from the figure, again supporting the attractor paradigm [25, 26]. We may speculate that the optimal region could be related to a region of maximal fitness for the tumor in the given tissue.

The intuition induces us to relate the tumor stage to the coordinate along the tumor axis PC1. The correspondence, however, is not exact. Although there seems to be a correlation between stage and mean displacement towards the tumor region, many tumors in the initial stages are already at the center of the cloud. This could be related to the fact that the observed distribution of samples is probably related to the fitness distribution and the transition region should be a low-fitness zone [33]. Despite the scarcity of tumor samples in the IV stage of LIHC and LUSC, a similar conclusion can be reached from the analysis of their corresponding stage evolution (see S4 Fig in S1 File).

## Statement 5

*There is a landscape of cancer in gene expression space with an approximate border separating normal tissues from tumors.*

We want to draw a picture in which both normal tissues and tumors in different localizations are represented. In a way, this is a picture involving tissue differentiation and cancer. It is not surprising that pan-cancer genes will play a role in both processes.

We shall use the normal, $\mathbf{e_{normal}}$, and tumor, $\mathbf{e_{tumor}}$, averages (geometric mean) of the gene expression vectors for each localization in order to reduce normal and tumor clouds to their respective normal and tumor centers. The common reference for all the tissues, $\mathbf{e_{ref}^{all}}$, is then computed as the average expression vector of the normal centers, i.e. the center of the cloud of normal centers. By using this reference, we can obtain the logarithmic fold variation, $\mathbf{e_{fold}^{tissue}}$, for

each tissue and build the covariance matrix of all the localizations. The dimension of this matrix is still equal to the number of genes, which can be reduced by the PCA processing in order to obtain a simplified description in 2–3 variables.

The first aspect to be stressed is that the first 2 PCs accounts only for 37% of the total data variance. The relative importance of these two variables is not so apparent as in the case of individual tissues. This is probably due to the big dispersion of the data for normal tissues, related to tissue differentiation, sometimes even larger than separations between a normal tissue and the respective tumor.

As a consequence of the dispersion of normal tissues, we do not have a "cancer axis" or direction, as in individual tissues. In order to draw a frontier between normal and tumor regions, we shall include higher PCs. The next component, PC3 accounts for 12% of the data variance.

We show in Fig 6 the (PC1, PC3) plane, which indeed suggests that there is a border. Actually, the regions and the border are high dimensional, but the 2D figure captures the essential features. We may baptize this figure as the "approximate normal versus cancer" or "tissue differentiation versus cancer" plane. It is apparent from the figure, that the transition from a

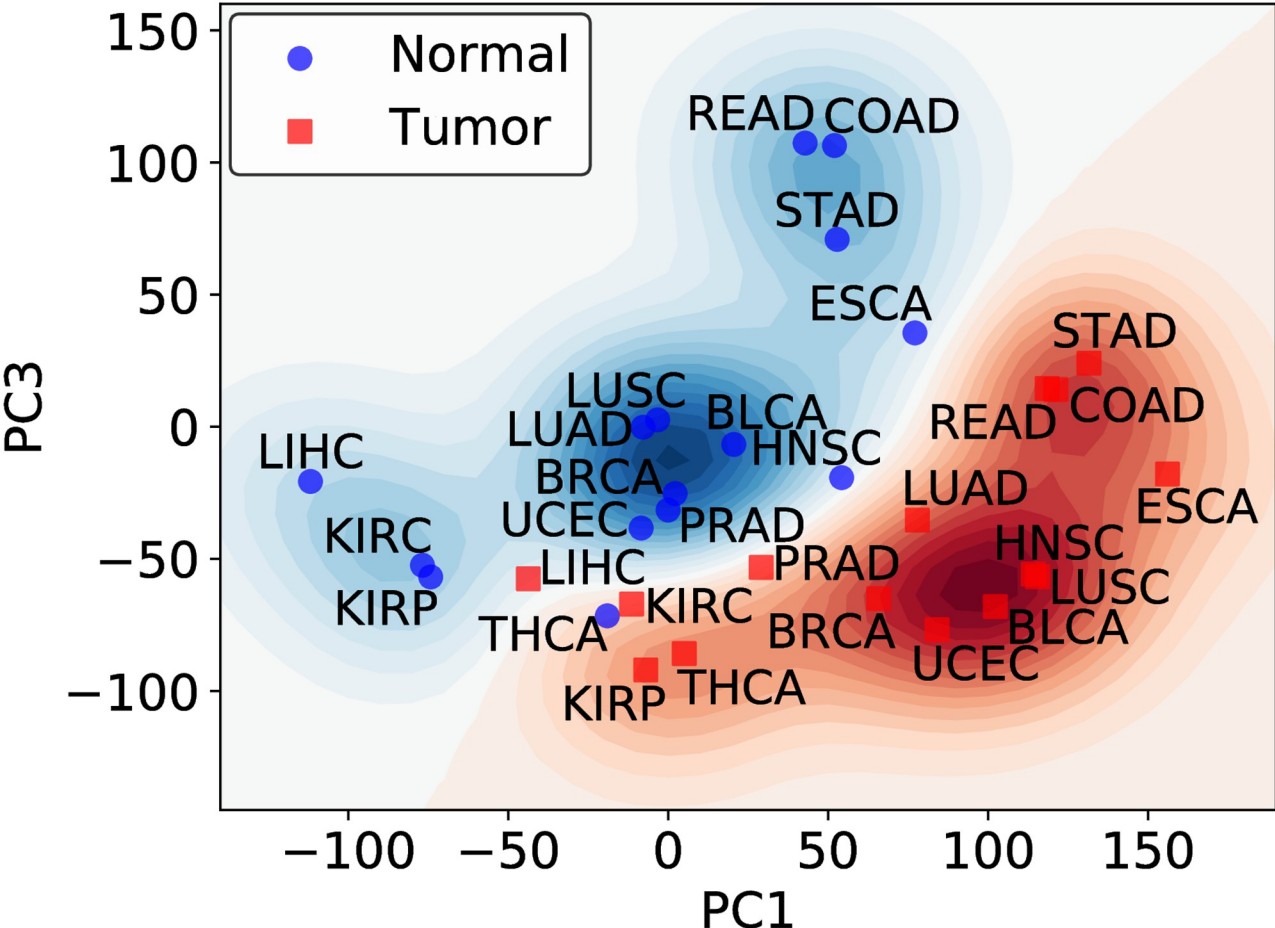

**Fig 6. The gene expression landscape in the (PC1, PC3) plane.** Each point in the diagram represents the average of samples in a given localization. For simplicity, normal tissues are labeled with the corresponding tumor indexes. The approximate border between the normal and tumor regions is apparent.

normal tissue to the corresponding tumor implies crossing the border, and involves simultaneous displacements along the PC1 and PC3 axes.

The unitary vectors along these axes allow the identification of genes with the highest weights. It is very interesting that pan-cancer genes are among the most important genes in these vectors. For example, ADH1B and UBE2C are included in the set of 8 most important genes along PC1: PI3 (+), ADH1B (-), MYBL2 (+), UBE2C (+), ALB (-), CEACAM5 (+), CST1 (+) and MMP1 (+).

A more detailed analysis of the border between normal tissues and tumors is needed, in particular a study of the role of pan-cancer genes. We notice that the separation between the normal and cancer manifolds takes place also in protein expression space [51]. Fig 6 provides also a view to groups or clusters of tumors, based on distances in GE space which, as mentioned above, are a true measure of closeness. A similar analysis is performed in S6 Fig in S1 File. It deserves however a further work.

## Discussion

Data processing techniques for dimensional reduction like PCA have proved to be a very useful tool for analyzing gene expression data. The current application to cancer allows to extract information contributing to the understanding of the cancer state from a theoretical point of view. In particular, the results of the PCA processing of the GE data for the 15 studied tumors support Kauffman's theory of cancer as an attractor state, i.e once a portion of the tissue escapes from the normal region it is driven to the cancer basin of attraction.

Already with the first component, PCA manages to separate normal and tumor data samples into two very disengaged groups for all studied tumor localizations. The PC variables, although of statistical origin, can be used to describe the state and evolution of the tissue. By computing the cumulative variance we show that the number of relevant PC coordinates for a tissue, i.e. the effective number of degrees of freedom of the biological system, do not seem to be much larger than 10, in spite of the huge number (more than 60000) of constituent genes.

The transition from a normal tissue to a tumor involves the modification of thousands of genes. These genes do not act independently, but in a concerted way. There is a gene expression profile for each tumor, which can be obtained from the unitary vector along PC1. The profile can be used to identify potential biomarkers and target genes.

Tumors in different localizations share hundreds or even thousands of genes from their profiles. The number of shared genes inversely correlates with the distance between the two tumors in GE space. We found 6 genes common to the 15 tumor localizations. These genes seem to play a significant role both in the process of tissue differentiation and in delimiting the border between the set of normal tissues and their corresponding tumors.

Our results are approximate and semi-quantitative, in the sense that they are limited by the number of samples and could be numerically modified if a larger data set becomes available, but at the same time they are simple, general and unbiased, in the sense that no modeling or elaborated mathematical treatments are used. We try to keep the interpretation of the results as devoid as possible of any speculation. The results of our paper raise some interesting questions as, for example, what would happen if we target a few of the most significant genes in the cancer profile? Could this kind of intervention induce a rearrangement of the whole profile preventing the tumor to evolve to more advanced stages? Questions are also raised in relation to the overall landscape in gene expression space shown in Fig 6. For example, could we use the pan-cancer genes (i.e, the genes common to all 15 tumors) in therapies designed to act in this group of tumors?

In the paper, we focused on qualitative aspects. However, there is the possibility to quantify and to model some aspects of cancer. For example, notice that from figures like Fig 1 we can estimate the dimensions of the normal and cancer regions in gene expression space and the distance between their centers. From this data, and the statement that progression is described by a single variable, we may devise a one dimensional model for tumorigenesis in a given tissue [35]. As mentioned above, the GE data allows also to measure the volumes of the basins of attraction in the normal and cancer states [33], to suggest a picture of smooth and abrupt transitions and the relation with GE rearrangements [34], etc. Work along these and other interesting directions is in progress.

## Methods

We analyze TCGA data for gene expression [11]. This is a well curated database. We selected the 15 cancer types shown in Table 1 on the basis of two conditions: i) The number of normal samples is greater than or equal to 10, and ii) The number of tumor samples is greater than or equal to 160.

The TCGA data involve measurements of the gene expressions for 60483 RNA- and protein-coding genes [52, 53]. This is the dimension of the matrices used in our principal component analysis. The data is in the FPKM format, corresponding to the number of fragments per kilo-base of gene length per mega-base of reads [54]. This is a way of normalizing measurements in a given sample.

On the other hand, in order to compute the average expression of a gene in a set of samples commonly the median or the geometric mean are used. We prefer geometric averages, but then the data should be slightly distorted to avoid zeroes. To this end, we added a constant 0.1 to the data. Indeed, we show in S5 Fig in S1 File expressions from a typical data file (PRAD case). Notice that there are around 28000 not transcribed genes (expression exactly zero), and only around 30000 genes with expression above 0.1. By applying this regularization procedure, the differential expression of genes with expression values below 0.1 is set to 1, and they will practically have zero contribution to the PCA. This is a very simple way of guaranteeing that only statistical significant genes enter the analysis.

For each cancer localization, we take the mean geometric average over normal samples in order to define the reference expression for each gene, $e_{ref}$. Then the normalized or differential expression is defined as: $e_{diff} = e/e_{ref}$. The logarithmic fold variation is defined in terms of the base 2 logarithm, $e_{fold} = \log_2(e_{diff})$. Besides reducing the variance, the logarithm allows treating over- and under-expression in a symmetrical way. In addition, we verified that the log transformation makes the data more close to a normal distribution, a requirement of the PCA method [27–29].

Deviations and variances are measured with respect to $e_{fold} = 0$. That is, with respect to the average over normal samples. This election is quite natural, because normal samples are the majority in a population, individuals with cancer are rare.

With these assumptions, the covariance matrix is written as

$$\sigma_{ij} = \sum \mathbf{e_{fold}}^i(s)\mathbf{e_{fold}}^j(s)/(N_{samples} - 1), \qquad (2)$$

where the sum runs over the samples, $s$, and $N_{samples}$ is the total number of samples (normal plus tumor). $\mathbf{e_{fold}}^i(s)$ is the fold variation of gene $i$ in sample $s$.

The matrix $\sigma$ is then diagonalized. As mentioned, its dimension is 60483. The obtained eigenvectors define the Principal Component (PC) axes: PC1, PC2, etc, and the projection over them define the new state variables. By definition, the index of the component is assigned

from the highest to the lowest fraction of the total variance captured by the PC in the sample set, thus the highest percentage of variance corresponds to the PC1 axis.

With this procedure, around 10 PCs are enough for an approximate description of the region of the gene expression space occupied by the set of samples. Thus, we need only a small number of the eigenvalues and eigenvectors of $\sigma$. To this end, we use a Lanczos routine in Python language, available on https://github.com/DarioALeonValido/evolp, and run it in a node with 2 processors, 12 cores and 64 GB of RAM memory. As a result, we get the first 100 eigenvalues and their corresponding eigenvectors.

## Supporting information

**S1 File. Appendixes with figures.** 1.1) Fraction of over- and under-expressed genes in the 15 tumors under study. 1.2) The contribution of genes to the unitary vector along PC1. 1.3) The proximity of tumors in GE space and the number of shared genes. 1.4) Stages in the evolution of tumors. 1.5) Pan-cancer genes and their characteristics. 1.6) Range of expression values in a typical TCGA data file. 1.7) Differentially expressed genes and top pathways. 1.8) Clustering analysis based on S2 Table.
(PDF)

**S1 Table. Pan-cancer genes and their characteristics.**
(XLS)

**S2 Table. Differentially expressed genes and top pathways.**
(XLS)

## Acknowledgments

A.G. acknowledges the Office of External Activities of the Abdus Salam Centre for Theoretical Physics and the University of Electronic Science and Technology of China for support. DA.L. acknowledges support from the Norwegian University of Life Sciences. The research is carried on under a project of the Platform for Bio-informatics of BioCubaFarma, Cuba. The data for the present analysis come from the TCGA Research Network [11].

## Author Contributions

**Conceptualization:** Augusto Gonzalez.

**Formal analysis:** Augusto Gonzalez, Dario A. Leon.

**Investigation:** Augusto Gonzalez, Dario A. Leon.

**Methodology:** Augusto Gonzalez, Dario A. Leon.

**Software:** Augusto Gonzalez, Dario A. Leon.

**Supervision:** Augusto Gonzalez.

**Validation:** Augusto Gonzalez, Dario A. Leon, Yasser Perera, Rolando Perez.

**Visualization:** Dario A. Leon.

**Writing – original draft:** Augusto Gonzalez.

**Writing – review & editing:** Augusto Gonzalez, Dario A. Leon, Yasser Perera, Rolando Perez.

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
