## [Decision Letter · Decision Letter 0]

20 Sep 2022

PONE-D-22-21174On the gene expression landscape of cancerPLOS ONE

Dear Dr. Dario Alejandro Leon Valido,

Thank you for submitting your manuscript to PLOS ONE. After careful consideration, we feel that it has merit but does not fully meet PLOS ONE’s publication criteria as it currently stands. Therefore, we invite you to submit a revised version of the manuscript that addresses the points raised (by Reviewers 1 and 2)  during the review process.

We look forward to receiving your revised manuscript.

Kind regards,

Elingarami Sauli, PhD

Academic Editor

PLOS ONE

Journal Requirements:

2. Please ensure that you have specified (1) whether consent was informed and (2) what type you obtained (for instance, written or verbal, and if verbal, how it was documented and witnessed). If your study included minors, state whether you obtained consent from parents or guardians. If the need for consent was waived by the ethics committee, please include this information.

4. Please update your submission to use the PLOS LaTeX template. The template and more information on our requirements for LaTeX submissions can be found at http://journals.plos.org/plosone/s/latex.

"The authors received no specific funding for this work."

"A.G. acknowledges the Office of External Activities of the Abdus Salam Centre for Theoretical Physics and the University of Electronic Science and Technology of China for support. DA.L. acknowledges financialn support from the Norwegian University of Live Sciences for publication fees. The research is carried on under a project of the Platform for Bio-informatics of BioCubaFarma, Cuba. The data for the present analysis come from the TCGA Research Network."

"The authors received no specific funding for this work."

7. Thank you for stating the following in your Competing Interests section:  

"The authors declare no compiting interest."

Reviewers' comments:

Reviewer's Responses to Questions

**Comments to the Author**

1. Is the manuscript technically sound, and do the data support the conclusions?

Reviewer #1: Yes

Reviewer #2: Yes

2. Has the statistical analysis been performed appropriately and rigorously? 

Reviewer #1: I Don't Know

Reviewer #2: Yes

3. Have the authors made all data underlying the findings in their manuscript fully available?

Reviewer #1: Yes

Reviewer #2: Yes

4. Is the manuscript presented in an intelligible fashion and written in standard English?

Reviewer #1: Yes

Reviewer #2: Yes

5. Review Comments to the Author

Reviewer #1: Review of “On the Gene Expression Landscape of Cancer” by Gonzales et al.

This manuscript is interesting and thought-provoking in that it addresses “grand” questions of cancer from a higher-order viewpoint, rather than ever narrower questions in ever more detail as most papers seem to do nowadays. However, the writing could be clearly improved and be more concise, precise and use a more scientific language. There are many “filling” words and sentences that can be omitted without losing information.

The following major and minor points should be addressed:

Major points

1. The first two paragraphs of the Results section could be considerably shortened. Parts of it should be transferred to the Methods section, and/or covered by references. It is not necessary to explain in this paper how RNA-seq works, and it is not part of this manuscript`s results. The authors state “In the Methods section, the used PCA methodology is briefly explained”. This statement is sufficient for the Results section, further details on their PCA methodology should be in the Methods.

2. Figure 2 and the corresponding Text shows “a few thousand of genes” differentially expressed between tumors and normal tissue, and the magnitude of this differential expression is at least two-fold, and up to >100-fold. This is unusual, since similar previous analyses show fewer differentially expressed genes, and a lower magnitude of the differences. Are all these differences statistically significant (by t-test or similar test statistics)? Have the authors properly normalized the gene expression data (per-gene and per-sample normalization)?

3. Legends to Figures and Tables should explain what is shown in corresponding Figures and Tables in more detail and with more precision, e.g. all abbreviations should be defined (or at least refer to an item where those are defined), and x- and y-axes should be well defined (e.g. x-axis in Figure 3 is labelled “Differential Expression” – are these fold-changes? Probably yes, but this should be defined!).

4. In statement 4, second part (page 6), the effect of tumor stage (I, II, II, IV) on gene expression of tumors is discussed, and it is assumed that higher-stage tumors should be progressed further on the normal-tumor gene expression axis. However, whereas expression is analysed in the primary tumor, the tumor stage is not a property of the primary tumor alone, e.g. stage IV commonly means that distant metastasis are present, and lymph node metastasis also play a key role in determining the tumor stage (which may or may not affect the gene expression state of the primary tumor analysed here). Probably the tumor size or tumor grade (both a property of the primary tumor) would be a better parameter than tumor stage to address this question. In any case, it should be defined (or a reference provided), how tumor stage I-IV are defined in KIRC, and this potential limitation should be discussed.

Minor points

1. “Statement 1-5” in Results: Are these results, conclusions or hypothesis? More precision in wording is needed.

2. Statement 1 (page 4): the authors refer to “the left panel” and “the right panel” of Figure 1. They probably mean the upper panel and the lower panel.

3. This reviewer recommends to omit the scale lines within the panels in Figure 2, they are distracting and may cause problems during printing and photo-copying.

4. Page 4 bottom: “the rest of the 60000 genes”. The authors should explain how they arrive at this number. There are about 25000 human protein-coding human genes, were additional genes (miRNAs, long nc RNAs etc.) included in this number? The authors should provide their (or the TCGA’s) definition of “gene”.

5. Page 6: “Well defined regions in expression space is a fact in favor of the attractor paradigm of cancer” could be replaced by “Well defined regions in expression space support the attractor paradigm of cancer”.

6. “sub-expression” should rather be termed “under-expression”, as opposed to over-expression, or “up-regulation” and “down-regulation” should be used (throughout the manuscript).

7. The Formatting of the References apparently does not adhere to PLOS One style, in particular, the full first names of authors are provided.

Reviewer #2: PCA has been used to analyze GE data from TCGA dataset. In detail, the results of the PCA of 15 different tumors seem to support 5 different facts or statements. Overall, the manuscript is well written and PCA well performed. I have some minor comments:

- As authors stated: "results are simple, general and unbiased, in the sense that no modeling or elaborated mathematical treatments are used". It would be better to describe the possible limits of this approach.

- For example, could we use the pan-cancer genes (i.e, the genes common to all 15 tumors) in therapies designed to act in this group of tumors? Are these druggable targets?

6. PLOS authors have the option to publish the peer review history of their article (what does this mean?). If published, this will include your full peer review and any attached files.

Reviewer #1: No

Reviewer #2: No

---

## [Author Response · Author response to Decision Letter 0]

24 Oct 2022

Reviewer 1:

This manuscript is interesting and thought-provoking in that it addresses “grand”

questions of cancer from a higher-order viewpoint, rather than ever narrower questions

in ever more detail as most papers seem to do nowadays. However, the writing could be

clearly improved and be more concise, precise and use a more scientific language. There are

many “filling” words and sentences that can be omitted without losing information.

Authors Reply:

We thank the reviewer for his/her careful reading of the manuscript and his/her positive

comments and suggestions.

We have rephrased some sentences along the hole manuscript and added more phrases

to make the text clearer and more compressible. All the changes are highlighted in the new

version.

Reviewer 1:

The following major and minor points should be addressed: Major points 1. The first

two paragraphs of the Results section could be considerably shortened. Parts of it should

be transferred to the Methods section, and/or covered by references. It is not necessary

to explain in this paper how RNA-seq works, and it is not part of this manuscript‘s

results. The authors state “In the Methods section, the used PCA methodology is briefly

explained”. This statement is sufficient for the Results section, further details on their

PCA methodology should be in the Methods.

Authors Reply:

We thanks the referee for his/her suggestions. We have shortened the two paragraphs

and moved not relevant information to the Method section.

Reviewer 1:

2. Figure 2 and the corresponding Text shows “a few thousand of genes” differentially

expressed between tumors and normal tissue, and the magnitude of this differential

expression is at least two-fold, and up to ¿100-fold. This is unusual, since similar previous

analyses show fewer differentially expressed genes, and a lower magnitude of the differences.

2

Are all these differences statistically significant (by t-test or similar test statistics)? Have

the authors properly normalized the gene expression data (per-gene and per-sample

normalization)?

Authors Reply:

We thank the referee for this comment. In the new version of the manuscript we explain

in more detail the normalization and the used methodology. Our simple procedure discards

statistically irrelevant genes (its differential expression becomes 1 and the log fold variation,

which enters the principal component analysis, becomes 0). Genes with high differential

expression, highlighted by the methodology, have always a biological meaning.

Here, we would like to show data for PRAD as an example. The 1st figure is similar

to Fig. 3 of our manuscript. There are only two genes (dlx1 and pca3) with differential

expression higher than 20. There are also around 1000 genes with edif f > 2, and around 5000

genes with edif f > 1.5. Note that a study of statistically significant differentially expressed

genes in PRAD identifies around 11,000 genes (doi: 10.1371/journal.pone.0145322).

In paper doi: 10.1038/s41598-021-87764-0, we relate this relative high GE rearrangement

(that is, thousands of diff expressed genes) to a discontinuous transition between the normal

and the tumor state.

The 2nd figure plots sample data for pca3 (a known PRAD marker) in age groups.

Mean geometric averages are represented in red. The NY group is taken as reference for

normalization. This 2nd fig contributes with 3 point to the 1st figure: that is, one for each

of the NO, TY and TO curves.

Reviewer 1:

3. Legends to Figures and Tables should explain what is shown in corresponding

Figures and Tables in more detail and with more precision, e.g. all abbreviations should

be defined (or at least refer to an item where those are defined), and x- and y-axes should

be well defined (e.g. x-axis in Figure 3 is labelled “Differential Expression” – are these

fold-changes? Probably yes, but this should be defined!).

Authors Reply:

We thanks the referee for his/her comments. We have added more details to the caption

3

FIG. 1: PRAD

FIG. 2: PRAD

4

of the figures and tables. In particular, the x axis in Fig. 3 is properly the differential

expression. The log-log plot is simply to show the Paretto law in the tail distribution

function.

Reviewer 1:

4. In statement 4, second part (page 6), the effect of tumor stage (I, II, II, IV) on gene

expression of tumors is discussed, and it is assumed that higher-stage tumors should be

progressed further on the normal-tumor gene expression axis. However, whereas expression

is analysed in the primary tumor, the tumor stage is not a property of the primary tumor

alone, e.g. stage IV commonly means that distant metastasis are present, and lymph node

metastasis also play a key role in determining the tumor stage (which may or may not affect

the gene expression state of the primary tumor analysed here). Probably the tumor size or

tumor grade (both a property of the primary tumor) would be a better parameter than tumor

stage to address this question. In any case, it should be defined (or a reference provided),

how tumor stage I-IV are defined in KIRC, and this potential limitation should be discussed.

Authors Reply:

We recognize the limitations to draw “attractor basins” even in the context of advanced

tumors stages (III-IV) by only surveying the primary tumor. However, our assumption

here is that in primary tumors some information remains in their gene expression profiles,

concerning the original seeding of the distant metastasis, and/or that the process of cell

invasion and dissemination from this primary site is continuous.

In the same line, other groups have also extracted and proposed prognostic signatures for

advanced KIRC stages by studying primary tumor samples. See, for example:

https://doi.org/10.3389/fbioe.2019.00270

https://doi.org/10.1111/cbdd.14141

https://doi.org/10.3389/fonc.2022.912155

Reviewer 1:

Minor points 1. “Statement 1-5” in Results: Are these results, conclusions or hypothesis?

More precision in wording is needed.

5

Authors Reply:

We have clarified at the beginning of the Results section, that the statements refer to

results/conclusions, following straightforwardly from the analysis of the GE data.

Reviewer 1:

2. Statement 1 (page 4): the authors refer to “the left panel” and “the right panel” of

Figure 1. They probably mean the upper panel and the lower panel.

Authors Reply:

We thanks the referee for pointing this out, we have corrected it.

Reviewer 1:

3. This reviewer recommends to omit the scale lines within the panels in Figure 2, they

are distracting and may cause problems during printing and photo-copying.

Authors Reply:

Thanks the reviever for this comment, we have removed the grids.

Reviewer 1:

4. Page 4 bottom: “the rest of the 60000 genes”. The authors should explain how they

arrive at this number. There are about 25000 human protein-coding human genes, were

additional genes (miRNAs, long nc RNAs etc.) included in this number? The authors

should provide their (or the TCGA’s) definition of “gene”.

Authors Reply:

The referee is right, TCGA data involves measurement of the expression for 60487 RNA-

and protein-coding genes in the Ensembl notation. We have made explicit this point in the

text.

Reviewer 1:

5. Page 6: “Well defined regions in expression space is a fact in favor of the attractor

paradigm of cancer” could be replaced by “Well defined regions in expression space support

6

the attractor paradigm of cancer”.

Authors Reply:

We thanks the referee for pointing this out, we have corrected it accordingly.

Reviewer 1:

6. “sub-expression” should rather be termed “under-expression”, as opposed to over-

expression, or “up-regulation” and “down-regulation” should be used (throughout the

manuscript).

Authors Reply:

We thanks the referee for pointing this out, we have uniformized the notation.

Reviewer 1:

7. The Formatting of the References apparently does not adhere to PLOS One style, in

particular, the full first names of authors are provided.

Authors Reply:

We thanks the referee for pointing this out, we have corrected the format of the references

and the manuscript accordingly.

Reviewer 2:

PCA has been used to analyze GE data from TCGA dataset. In detail, the results of

the PCA of 15 different tumors seem to support 5 different facts or statements. Overall,

the manuscript is well written and PCA well performed.

Authors Reply:

We thanks the referee for his/her very positive comments.

Reviewer 2:

I have some minor comments: - As authors stated: ”results are simple, general and

unbiased, in the sense that no modeling or elaborated mathematical treatments are used”.

7

It would be better to describe the possible limits of this approach. - For example, could we

use the pan-cancer genes (i.e, the genes common to all 15 tumors) in therapies designed to

act in this group of tumors? Are these druggable targets?

Authors Reply:

We thanks the referee for raising these points. We have stressed in the new version of

the manuscript that the results of our PCA analysis are limited mainly by the number of

available samples. The second raised question on the pan-cancer genes is really interesting,

but beyond the scope of the paper.

Druggable targets are defined as those molecules (i.e. proteins, peptides, nucleic acids)

which levels and/or specific activities can by modulated by a drug, which can consist of

a small molecular weight chemical compound (SMOL) or a biologic (BIOL), such as an

antibody or a recombinant protein. Two issues need to be considered on the putative

druggability of these reported pan-cancer genes: 1) Gene expression changes highlighted here

need to be corroborated at the protein level if we attempt to develop a target therapy against

the final protein product. 2) Usually, protein down-regulation and/or diminished activity,

are harder to tackle than protein over-expression and/or activity inhibition. However, as

described below, there are experimental and approved drugs for 2/3 up-regulated pan-cancer

genes, whereas there are recent reported drugs aiming at the three down-regulated genes:

a) MMP11 (Up, Matrix Metallopeptidase 11): at least two experimental drugs are re-

ported at https://www.genecards.org/cgi-bin/carddisp.pl?gene=MMP11

b) C7 (Down, Complement Component C7): at least one experimental drug is reported

at https://www.genecards.org/cgi-bin/carddisp.pl?gene=C7&keywords=C7

c) ANGPTL1 (Down,Angiopoietin Like 1): one inferred drug is reported at

https://www.genecards.org/cgi-bin/carddisp.pl?gene=ANGPTL1&keywords=

ANGPTL1#drugs_compounds

d) UBE2C (Up, Ubiquitin Conjugating Enzyme E2 C): at least three approved drugs are

reported at https://www.genecards.org/cgi-bin/carddisp.pl?gene=UBE2C&keywords=

UBE2C#drugs_compounds

e) IQGAP3 (Up, IQ Motif Containing GTPase Activating Protein 3): no reported or in-

ferred drug so far at https://www.genecards.org/cgi-bin/carddisp.pl?gene=IQGAP3&

keywords=IQGAP3

8

f) ADH1B (Down, Alcohol Dehydrogenase 1B (Class I), Beta Polypeptide): there are

more than 5 approved drugs and another handful under developmental stages.

Of note, in a former comprehensive redefinition of the druggable genome (doi: 10.1126/sc-

itranslmed.aag1166), the authors stratified the druggable gene set into three tiers corre-

sponding to position in the drug development pipeline. Tier 1 (1427 genes) included efficacy

targets of approved small molecules and bio-therapeutic drugs as well as clinical-phase drug

candidates. Tier 2 was composed of 682 genes encoding targets with known bioactive drug-

like small-molecule binding partners as well as those with ≥ 50% identity (over ≥ 75% of

the sequence) with approved drug targets. Tier 3 contained 2370 genes encoding secreted or

extracellular proteins, proteins with more distant similarity to approved drug targets, and

members of key druggable gene families not already included in tier 1 or 2. Furthermore, the

more frequent Pfam-A domain content in three tiers of druggable genes were also described.

After inspecting this data we noted that ADH1B was positioned into Tier 1, whereas

MMP11, C7 and ANGPTL1 were located to Tier 3B. Overall, only IQGAP3 remains elusive

as a putative druggable target within the six pan-cancer genes. IQGAP3 contains the

Rho GTPase activation prot domains not included in tiers 1-3.

---

## [Decision Letter · Decision Letter 1]

3 Nov 2022

On the gene expression landscape of cancer

PONE-D-22-21174R1

Dear Dr. Dario Alejandro Leon Valido,

We’re pleased to inform you that your manuscript has been judged scientifically suitable for publication and will be formally accepted for publication once it meets all outstanding technical requirements.

Kind regards,

Elingarami Sauli, PhD

Academic Editor

PLOS ONE

Additional Editor Comments (optional):

Reviewers' comments:

Reviewer's Responses to Questions

**Comments to the Author**

1. If the authors have adequately addressed your comments raised in a previous round of review and you feel that this manuscript is now acceptable for publication, you may indicate that here to bypass the “Comments to the Author” section, enter your conflict of interest statement in the “Confidential to Editor” section, and submit your "Accept" recommendation.

Reviewer #2: All comments have been addressed

2. Is the manuscript technically sound, and do the data support the conclusions?

Reviewer #2: Yes

3. Has the statistical analysis been performed appropriately and rigorously? 

Reviewer #2: Yes

4. Have the authors made all data underlying the findings in their manuscript fully available?

Reviewer #2: Yes

5. Is the manuscript presented in an intelligible fashion and written in standard English?

Reviewer #2: Yes

6. Review Comments to the Author

Reviewer #2: My comments have been all addressed and the manuscript improved. I suggest accepting the manuscript.

7. PLOS authors have the option to publish the peer review history of their article (what does this mean?). If published, this will include your full peer review and any attached files.

Reviewer #2: No

---

## [Editor Report · Acceptance letter]

11 Nov 2022

PONE-D-22-21174R1 

On the gene expression landscape of cancer 

Dear Dr. Leon Valido:

I'm pleased to inform you that your manuscript has been deemed suitable for publication in PLOS ONE. Congratulations! Your manuscript is now with our production department. 

Kind regards, 

on behalf of

Dr. Elingarami Sauli 

Academic Editor

PLOS ONE